# A Study of Movement Classification of the Lower Limb Based on up to 4-EMG Channels

**Diana C. Toledo-Pérez** [1,†]**, Miguel A. Martínez-Prado** [2,†]**, Roberto A. Gómez-Loenzo** [2,†]**, Wilfrido J. Paredes-García** [2,†] **and Juvenal Rodríguez-Reséndiz** [2,*]

1   División de Investigación y Posgrado, Facultad de Informática, Universidad Autónoma de Querétaro (UAQ), Av. de las Ciencias S/N, Juriquilla, Querétaro C.P. 76230, Mexico; dtoledo16@alumnos.uaq.mx
2   División de Investigación y Posgrado, Facultad de Ingeniería, Universidad Autónoma de Querétaro (UAQ), Cerro de las Campanas, S/N, Col. Las Campanas, Querétaro C.P. 76010, Mexico; miguel.prado@uaq.mx (M.A.M.-P.); rob@uaq.mx (R.A.G.-L.); wparedes17@alumnos.uaq.mx (W.J.P.-G.)
*   Correspondence: juvenal@uaq.edu.mx; Tel.: +52-442-192-1200
†   These authors contributed equally to this work.

**Abstract:** The number and position of sEMG electrodes have been studied extensively due to the need to improve the accuracy of the classification they carry out of the intention of movement. Nevertheless, increasing the number of channels used for this classification often increases their processing time as well. This research work contributes with a comparison of the classification accuracy based on the different number of sEMG signal channels (one to four) placed in the right lower limb of healthy subjects. The analysis is performed using Mean Absolute Values, Zero Crossings, Waveform Length, and Slope Sign Changes; these characteristics comprise the feature vector. The algorithm used for the classification is the Support Vector Machine after applying a Principal Component Analysis to the features. The results show that it is possible to reach more than 90% of classification accuracy by using 4 or 3 channels. Moreover, the difference obtained with 500 and 1000 samples, with 2, 3 and 4 channels, is not higher than 5%, which means that increasing the number of channels does not guarantee 100% precision in the classification.

**Keywords:** intention of movement classification; EMG-Signals; Support Vector Machines

## 1. Introduction

In recent decades, the use of signals obtained from the muscles has become popular due to its implementation in different applications such as health monitoring, assistive technology, and prosthetic control. This is due to the increase in technological advances in wearable electronics for the exploration of muscle signals.

When a muscle contraction or relaxation occurs, it generates an electrical potential that can be measured with an electromyographic sensor. There are two different approaches to place this kind of sensor—invasive and non-invasive methods. In the case of the former, the sensor is intramuscular; whereas in the latter, commonly called surface electromyography (sEMG), the sensor is placed on the skin surface; the former approach is the most common technique since it does not require surgical intervention.

To improve the classification accuracy, Oskoei, M. A. and Hu, H. [1] experimented with the quantity and type of characteristics used in the feature vector, while She et al. [2] varied the kernels utilized by the classifier.

Using another approach, Englehart K. and Hudgins B. [3] compared the effect in the accuracy due to the method used to obtain the features in frequency time, like Fast Fourier Transform, Wavelet Transform, and Wavelet Packet Transform. In general, the most common way used to improve this

accuracy is to find some algorithm, variation or combination of these, to try to reach an accuracy of 100% [1,4–13].

Some researchers have also increased the number of channels used to classify; for example, Fukuda O. et al. [9] used six sensors, authors [14–17] used eight and Ceseracciu et al. [18] even used sixteen, but, none yields 100% accuracy in the classifications. In an effort to improve accuracy, some researchers have not only increased the number of channels but also the number of features employed; for example, Alizadeh et al. [19] increased both the number of features up to 28 and the EMG channels up to six.

The methods for signal analysis involve time-domain and frequency-domain features, time-frequency analysis methods, power spectrum density, and higher-order spectra [20]. For example, Pancholi S. and Joshi A. M. [21] combines two of them, time and frequency domain features, using nine features for the time domain and seven features for the frequency domain, that is 16 in total. In order to analyze the sEMG signals, this work only considers time-domain features, since they are easy to compute and do not require any transformation. Therefore, Mean Absolute Value (MAV), Zero Crossings (ZC), Waveform Length (WL) and Slope Sign Changes (SSC) are recommended characteristics to obtain a better classifier performance [2,3,8,13,16,22].

Increasing the number of channels for the classification introduces a dimensionality problem, which leads to lower classification performance [23]. Some tools can be used to analyze signals to improve the classification accuracy without increasing the number of them processed, e.g., the Empirical Mode Decomposition used only in a single-channel [24].

Aside from the techniques looking for ways to improve the accuracy in classification, other research works are focused on reducing the dimensionality problem such as Principal Component Analysis (PCA), Independent Component Analysis (ICA), Linear Discriminant Analysis (LDA), Canonical Correlation Analysis (CCA), among others.

Although the goal of PCA is usually to find out an optimal linear transformation which represents the original data and to reduce the dimensionality of the features vector [25–28], this research work only uses this method in order to achieve better accuracy, not to reduce the dimensionality of features.

Support Vector Machines (SVMs) are used for classification because they have a high potential for classifying signals in myoelectric control systems since they can recognize complex patterns [1].

However, in previous research, the difference in classification accuracy caused by increasing the number of channels or by varying the muscle from which the EMG signal is extracted has not been shown. This study offers the researchers the opportunity to decide whether the increase in resources used for processing is worthwhile or not.

In this article, sEMG signals were recorded on four opposite muscles on the lower limb and are used to compare the classification accuracy; there were four different stages with an increase in the number of signals in each stage that is, in a first step, only one signal was used, then two of them, then three and finally four signals in a final stage. The muscles selected to place the sensors on them were tibials anterioris (TA), gastrocnemius medials (GM), biceps femoris (BF) and vastus lateralis (VL), which presents a better movement signal [29].

This paper is organized as follows. Section 2 provides a brief background of the conventional techniques used for sEMG signals analysis and the most commonly used features, and describes the Support Vector Machine algorithm and PCA. Section 3 describes the experimental design and the analysis of sEMG signals. Comparison results from the SVM classifier varying the number of channels and their origin are presented in Section 4. Section 5 presents our concluding remarks.

## 2. Background

### 2.1. Analysis of sEMG Signals

Myoelectric control success depends highly on the classification accuracy. Classification methods and feature extraction are essential to attain high performance in the classification for pattern recognition [1].

Depending on the level of muscle contraction, sEMG signals vary in amplitude, variance, energy, and frequency. Given those measures, a variety of features is extracted from them for their analysis. As mentioned earlier, the most recommended in literature are MAV, ZC, SSC, and WL, and are described in the following paragraphs.

- MAV: It is the average of the $N$ absolute values of the sEMG samples within a given time epoch, and is given by:

$$\text{MAV} = \frac{1}{N} \sum_{i=1}^{N} |x_i|. \tag{1}$$

- ZC: It is the number of times that the signal samples $\{x_i\}$ cross zero, whether it goes from a negative value to a positive one or the other way around, as in equation:

$$\text{ZC} = \sum_i f_{\text{ZC}}(x_i), \tag{2}$$

where

$$f_{\text{ZC}}(x_i) = \begin{cases} 1, & \text{if } x_i > 0 \quad \text{and} \quad x_{i+1} < 0 \\ & \text{or } x_i < 0 \quad \text{and} \quad x_{i+1} > 0, \\ 0, & \text{otherwise.} \end{cases} \tag{3}$$

- WL: It is the accumulated variation of a signal that can indicate the degree of signal oscillation and is given by equation:

$$\text{WL} = \sum_{i=1}^{N-1} |x_{i+1} - x_i|. \tag{4}$$

- SSC: It counts the number of times that the slope of the signal changes sign, which make necessary to evaluate where it is, where it was and where the signal goes. SSC is calculated with equation:

$$\text{SSC} = \sum_i f_{\text{SSC}}(x_i), \tag{5}$$

where

$$f_{\text{SSC}}(x_i) = \begin{cases} 1, & \text{if } x_i < x_{i+1} \quad \text{and} \quad x_i < x_{i-1} \\ & \text{or } x_i < x_{i+1} \quad \text{and} \quad x_i > x_{i-1}, \\ 0, & \text{otherwise.} \end{cases} \tag{6}$$

### 2.2. Principal Component Analysis

PCA is a statistical technique that performs a linear transformation from an original set of values into a smaller one of uncorrelated variables, which represents the most relevant information of the original set. Thus, the dimensionality of the original set is reduced or kept but never increased. The idea was conceived by K. Pearson [30] and later developed by Hotelling [31].

The PCA technique uses the covariance matrix from the original set ($X$) and the correlation between every one of these components, in such a way that a smaller $Y$ output space is found, by representing the statistical information contained in $X$ as it is described in equation:

$$Y = XC, \tag{7}$$

where $C$ is the $m \times n$ matrix with the principal components selected, where $n < m$, which implies the dimensionality reduction from the original set. The procedure to determine $C$ consists in constructing the covariance matrix, then compute the eigenvalues and eigenvectors to project the data matrix with these eigenvectors in decreasing magnitude order. Finally, it is only necessary to consider the desired information and to select the number of vectors that compose it.

*2.3. Support Vector Machines*

SVMs are commonly used as a classification algorithm for body movements, images, sounds, and other data. An SVM builds an optimum separation hyperplane in a feature space which is said to be of high dimension when the inputs are mapped using non-linear functions, to be able to distinguish between two or more object types. In 1995 this theory was introduced in [32].

In an SVM, the training algorithm is reformulated as a global and unique problem to solve using Quadratic Programming (QP) for input training data $(x_1, y_1), \ldots, (x_m, y_m) \in \mathbb{R}^N \times \{-1, +1\}$, where $x_i$ corresponds to the input value and $y_i$ the assigned value of the object type to which it belongs (also known as a class); if these data are not linearly separable, they are a mapped (non-linearly) by a kernel function $\varphi \colon \mathbb{R}^N \mapsto F$ into a characteristic space $F$. In this way, the obtained linear hyperplanes that separate the object types can be described as:

$$\omega \in \{x \mid \varphi(x) + b = 0\}, \qquad \omega \in \mathbb{R}^N, \quad b \in \mathbb{R} \tag{8}$$

Thus, by constructing an optimal hyperplane with the maximum value of the separation margin and a closed error $\xi$ in the training of the algorithm, the QP problem is stated as:

$$\min_{w,b} \frac{1}{2} \|\omega\|^2 + C \sum_{i=1}^{m} \xi_i \tag{9}$$

The first term in a cost function that generates a maximum separation margin between classes, while the second one provides an upper bound for mistakes in the training data. Finally, the constant $C \in [0, \infty)$ creates a compensation between the number of poorly classified samples with a maximum margin.

Finally, the solution to the objective function proposed in Equation (9) can be obtained as mentioned in the previous paragraph using QP tools or methods such as those proposed by Pérez-Hernández [33].

## 3. Methods and Experimentation

*3.1. Data Acquisition*

The data in this research were acquired from eight healthy subjects, four females and four males. All subjects are aged between 23 and 32 years old, normally limbed and without muscle disorders.

The sensor system was placed on the skin over the muscles and comprised nine electrodes, eight of them, positioned in pairs, sensing the differential potential from muscles, and the last one used as a ground reference. Electrodes used were *Kendall Medi Trace 200* (Ag/AgCI circular bipolar electrodes, with 10 mm in diameter with an adhesive conducting gel). The sEMG signals were amplified almost 1000 times after passing through the INA114 amplifier. Then, the signals went through an analog 60 Hz notch filter to remove electric line interference, implemented with an operational amplifier. Later, an offset was applied to the signal to set a reference voltage of 1.67 V, because the ADC (Analog to Digital Converter) has a range from 0 to 3.3 V (Figure 1).

The signals were sampled with the aid of an STM32F103C8 microcontroller with a 12-bit ADC at a sampling frequency of 1000 Hz; each sample was packed as 2 bytes, which were sent to a PC, and then stored in an ASCII text format.

Each pair of sensors was placed according to the distances described in [29] with a 2.5 cm separation between them to obtain the best signal quality: For VL the best place is at 66% of the muscle length on the line from the anterior spina iliaca superior to the lateral side of the patella; for the TA is at 47.5% on the line between the tip of the fibula and the tip of the medial malleolus. The optimal electrode position in GM is at 38% of the muscle length from the medial side of the popliteal cavity to the medial side of the Achilles tendon insertion, starting from the Achilles tendon; and, for BF 50%, the position on the line between the ischial tuberosity and the lateral epicondyle of the tibia presents

the best quality of the signal. For a thorough discussion of the relevant issues regarding electrode placement, refer to [34].

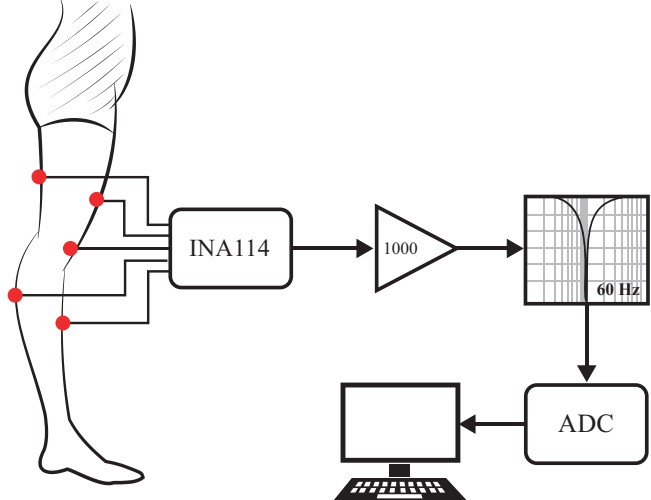

**Figure 1.** Basic experiment diagram.

### 3.2. Data Processing

For the data processing, MATLAB and the LIBSVM library version 3.2 were used in this work [35]. Although this library provides a module dedicated to applying a variety of kernels, such as linear, polynomial, RBF (Radial Basis Function), or sigmoid, none of them was required.

In software, two different digital filters were applied to remove undesirable noise from the collected sEMG signals. First a 60 Hz notch filter and then an elliptical bandpass filter between 10 and 500 Hz. The functions used were *filter*, *ellipord*, and *ellip*.

For the training process, first, the feature vector was built with MAV, ZC, WL, and SSC for windows of different sizes and for each of the channels individually; this was to make a comparison of the results with different schemes. In a second step, because of the differences between each feature the feature matrix was standardized. As a third step, a PCA analysis was performed without removing vectors from the transformation matrix. Finally, the obtained PCA matrix was multiplied by the feature matrix and the resulting matrix was used as input to train the SVM classifier.

Steps one and two were repeated for the test data set; later, the feature matrix obtained was multiplied by the PCA matrix and the resulting matrix was used to test the SVM.

### 3.3. Experimentation

Six classes of foot movement plus rest were considered for the research: lift the toe (LP), lift the heel (LT), move the toe to the right (PD), move the toe to the left (PI), lean on the heel (AT), lean on the toe (AP), and rest foot (RR). In the experiment, the subjects were sitting and started from a relaxation state and then performed the movement and held it for 5 s, and then they returned to the relaxation position. The movements were repeated 20 times with a resting period of 25 s between the movements by each subject. Tests were done in a single session.

The first window size considered was 250 ms, since 300 ms is an acceptable delay from the system in case that the intended use the system is controlling a prosthesis [3]. Also, considering other possible usages, another two window sizes were considered, namely, 500 and 1000 ms.

Finally, the collected data were divided into two groups, the training, and the testing data; ten samples for each group. In other words, the database is composed of 1120 movements, from eight different people (four females and four males) and seven different movements. Of these movements, 560 were used to train the SVM and the other 560 were used to test the classification accuracy.

## 4. Results

The obtained results are shown in Tables 1 and 2. The first table shows the best results in accuracy for each window size, considering one, two or three channels; with an additional row with the values for four channels. The first column contains the number of channels considered, and the last one has the channels with which the result was obtained.

**Table 1.** Best accuracy results obtained among the eight subjects.

| Number | Samples | | | Channels |
|---|---|---|---|---|
| | 250 | 500 | 1000 | |
| 1 | 90.00% | 91.43% | 95.71% | VL |
| 2 | 95.71% | 97.14% | 97.14% | GM & VL |
| 3 | 95.71% | 100.00% | 98.57% | TA, GM & VL |
| 4 | 95.71% | 100.00% | 100.00% | TA, GM, BF & VL |

**Table 2.** Results obtained with the lowest accuracy among the eight subjects.

| Number | Samples | | | Channels |
|---|---|---|---|---|
| | 250 | 500 | 1000 | |
| 1 | 52.86% | 55.71% | 64.29% | TA |
| 2 | 70.00% | 72.86% | 75.71% | TA & VL |
| 3 | 78.57% | 78.57% | 87.14% | TA, GM & BF |
| 4 | 81.43% | 81.43% | 87.14% | TA, GM, BF & VL |

Data shown in Table 1 indicates that the best muscle to extract movement information is *VL* since it appears with one, two or three channels; and the second-best option is *gastroctemius medialis*, also appearing with two or three channels. Additionally, the difference in the accuracy obtained with 500 and 1000 samples using two, three and four channels is of one sample at the most.

The results in Table 2 are the lowest scores, and these in turn show that *tibialis anterior* has not enough information to make a good classification, even if it is combined with the VL muscle. Also, the combination of three channels without the VL muscle has the worst performance. The accuracy of the classification increases less by increasing the window size than by increasing the number of channels.

Figure 2 shows a graphic with the average of the results obtained with a single channel, where the VL muscle presents the best accuracy classification and the TA muscle the worst. Additionally, the results of varying the sampling window size are not conclusive enough to state the recommended size.

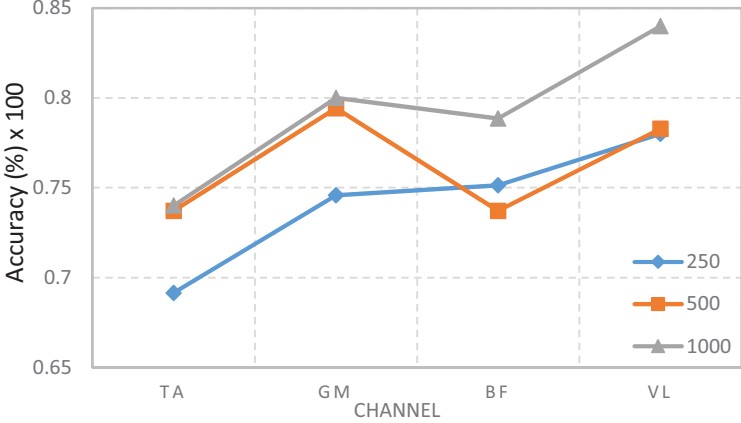

**Figure 2.** Classification accuracy with a single channel.

As shown in the comparison of two channels in Figure 3 the GM and VL muscles have a better performance than the rest. Furthermore, a more consistent performance can be achieved with a sampling rate of 1000 than with any other number of samples, but the difference with 500 is minimal in most cases. Figure 4 shows that the combination with GM, BF, and VL is better for classification than those that include channel TA; again, the difference between 500 and 1000 samples is minimal.

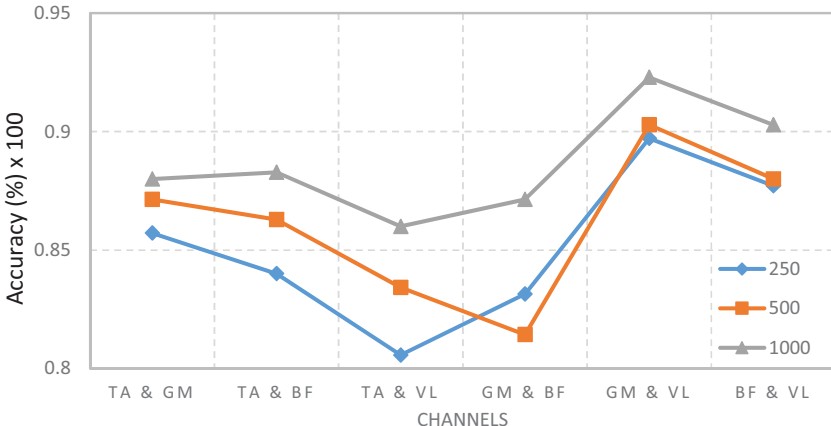

**Figure 3.** Classification accuracy with two channels.

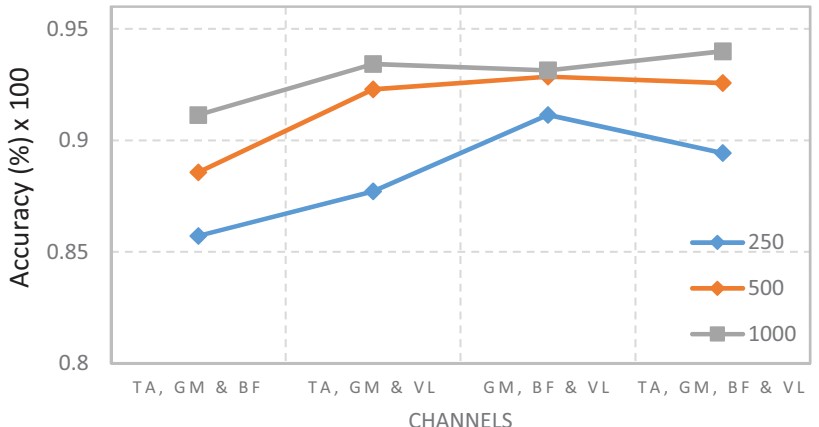

**Figure 4.** Classification accuracy with three and four channels.

In addition to the tables and graphics with accuracy scores, a channel forward selection of variables was also made based on the area of ROC (Receiver Operating Characteristic) curve multi-class and a classification error rate. The results obtained and their corresponding 95% confidence intervals with a sample size of 24 are shown in Table 3.

**Table 3.** Results of Channel forward selection of variables based on the area of ROC curve multi-class.

| Step | Selection | ROC Area | ROC Area CI | C. E. | C. E. CI |
|------|-----------|----------|-------------|-------|----------|
| 1 | Channel VL | 0.9397 | (0.8770, 1.00) | 0.1952 | (0.0224, 0.3680) |
| 2 | Channel GM & VL | 0.9517 | (0.8675, 1.00) | 0.1000 | (0.00, 0.2152) |
| 3 | Channel TA, GM & VL | 0.9673 | (0.9147, 1.00) | 0.0839 | (0.00, 0.1925) |
| 4 | All Channels | 0.9866 | (0.9426, 1.00) | 0.0506 | (0.00, 0.1604) |

Table 3 shows that there is no statistical evidence to affirm that the increase of channels offers an improvement in the quality indicators of the classification. Similarly, there is also not enough statistical evidence either to assert that a lower quantity channels proves beneficial. Subsequently,

the window size effect was analyzed in a fixed channel selection. Evidence that this does impact classifier quality indicators is illustrated in Figure 5.

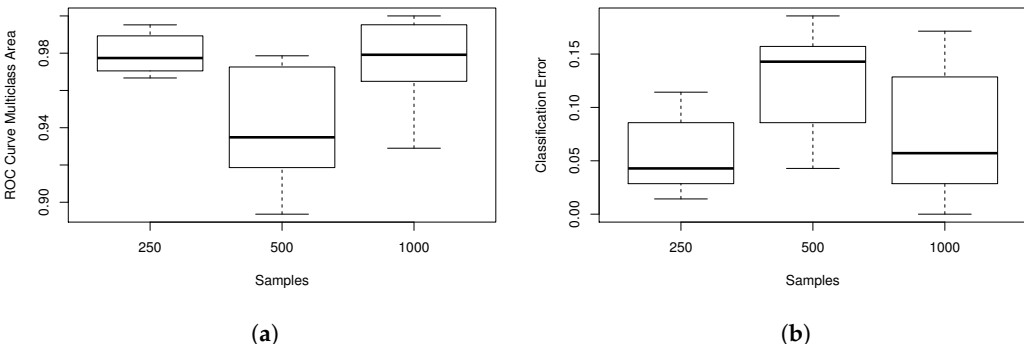

(**a**)  (**b**)

**Figure 5.** (**a**) Area under the curve estimation of ROC curve multi-class for different window sizes using all channels. (**b**) Error classification estimation for different window sizes using all channels.

Moreover, an ANalysis Of VAriance (ANOVA) was carried out to perform the hypothesis testing, and the obtained *p*-values were 0.0402 and 0.00768 for the effect of the area under the curve for the ROC curve multi-class and the error classification, respectively.

Furthermore, Figure 5 shows that a 250- or 1000-sample window size has a similar accuracy classification, i.e., the percentage of true positives increases in relation to the number of true positives and false positive resulting a positive effect. Furthermore, this same quantity also increases in comparison with the sum of false negative cases with true positive cases.

## 5. Discussion

In the first stage, the experiments were developed only with four subjects, three women and one man, the other three men and one woman were added in the second stage. We found a trend, that is, in the second stage we also obtained that the muscles individually analyzed, the one that obtained the least accuracy for the classification was the TA and the one with the highest precision was the VL.

The TA muscle (TA) presents the worst results when analyzed individually or jointly; this is probably because this is the muscle responsible for the dorsiflexion and inversion of the ankle, which helps the stabilization of the ankle during gait, so the selected movements do not require much of it. However, 100% of the classification accuracy was only obtained when this muscle was taken into account. However, it is also a muscle with a relatively small volume, compared to the others; this implies that the potential differential generated at the moment of movement is more difficult to measure. The muscle that offered the highest precision was the VL muscle.

On the other hand, it was expected that the difference in the accuracy of the classification, when increasing more channels, was significantly higher; however, the better results with two and three channels were similar to four channels, with the biggest difference being in the number of samples selected. In this sense, it was observed that when duplicating the number of samples, from 500 to 1000, the difference was not higher than 5% in most cases, so it is considered that it is not necessary to have such a large window size.

## 6. Conclusions

The obtained results with four channels were better than those with one single channel, but the difference with two and three channels is negligible. Even with 250 sample size, the results in three channels were better on average compared with four channels. The muscle with the worst performance was the TA. Additionally, the best results are obtained by taking the signal of opposing muscles. Finally,

this work aims to help the researcher decide how necessary it is to increase the resources used in the classification process to obtain the accuracy that is required.

Nevertheless, owing to the observed response variability presents a reduction as the number of channels increases, it is recommended employ a high number of channels to avoid changes in the classification by factors of sample size or subject. However, by considering just two channels, it is possible to achieve the same accuracy by making some adjustments to the classification algorithm.

**Author Contributions:** Conceptualization, D.C.T.-P. and J.R.-R.; Methodology, D.C.T.-P.; Software, D.C.T.-P.; Validation, D.C.T.-P., W.J.P.-G., and M.A.M.-P.; Formal analysis, D.C.T.-P.; Investigation and Visualization, D.C.T.-P., W.J.P.-G. and J.R.-R.; Data curation, D.C.T.-P., W.J.P.-G., and M.A.M.-P.; Writing—original draft preparation; Writing—original draft, review & editing, all the authors.

**Funding:** This research was funded by the "Consejo Nacional de Ciencia y Tecnología (CONACYT)" under the scholarship 561144.

**Acknowledgments:** We would like to thank the Graduate Studies Division from the Faculty of Computing at Universidad Autónoma de Querétaro by allowing me to make Ph.D. studies.

**Conflicts of Interest:** The authors declare no conflict of interest.

## Abbreviations

The following abbreviations are used in this manuscript:

| | |
|---|---|
| EMG | Electromyography |
| MAV | Mean Absolute Value |
| ZC | Zero Crossings |
| WL | Waveform Length |
| SSC | Slope Sign Changes |
| SVM | Support Vector Machine |
| PCA | Principal Component Analysis |
| TA | Tibialis Anterioris |
| GM | Gastroctemius Medials |
| BF | Biceps Femoris |
| VL | Vastus Lateralis |
| QP | Quadratic Programming |
| ADC | Analog Digital Converter |
| LP | Lift the toe |
| LT | Lift the heel |
| PD | Toe to the right |
| PI | Toe to the left |
| AT | Recharge on the heel |
| AP | Recharge on the toe |
| RR | Rest of the foot |
| ROC | Receiver Operating Characteristic |
| ANOVA | ANalysis Of VAriance |

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
