# Peer review of "A Study of Movement Classification of the Lower Limb Based on up to 4-EMG Channels"

_electronics, doi:10.3390/electronics8030259_

Round 1

Reviewer 1 Report

Manuscript ID: electronics-439471

General comments

The Manuscript ID: electronics-439471 presents some major weaknesses and many minor points mostly related to language.

The overall objecives are not clear. While classification of hand movements based on EMG may be important for prosthesis control, why is classification of lower limb movement important?

The purpose of the work is to test a classification algorithm that would identify lower limb movements based on a small number of channels of sEMG.  The major objection is that the features of any sEMG channel strongly depend on electrode size, interelectrode distance and electrode location on a muscle.

These geometrical parameters a not (or very poorly reported). The issue of repeatability of the classification results when the electrodes are repositioned is not addressed. Should the system be retrained every time it is used? This is not mentioned.

The authors are referred to the books

1.                  Hermens H., Freriks B, Merletti R., Stegeman D., Blok J., Rau G., Disselhorst-Klug C., Hagg G., European Recommendations for Surface Electromyography, RRD publish. ISBN 90-75452-15-2, 1999.

2.                  Barbero M., Rainoldi A, Merletti R. Atlas of muscle innervation zones: understanding surface EMG and its applications, Springer, Italy 2012

3.                  Merletti R, Farina D. (edts) Surface Electromyography: physiology, engineering and applications,  IEEE Press / J Wiley, USA, May 2016

And to the article :

Afsharipour B., Soedirdjo S Merletti R  Two dimensional surface EMG: the effect of electrode size, interelectrode distance and image truncation. BSPC 2019; 49, 298-307

In particular,  Fig 4 of the latter paper shows the low pass filtering effect of electrode size  and Fig. 9 shows the combined effect of size and interelectrode distance. Both affect ALL the sEMG features selected by the authors. A separation of 25 mm between two electrodes of one pair would create multiple dips in  the differential sEMG signals  (see Fig 9 of the paper mentioned above or fg 2.5 page 40 of  the third book indicated above. How were the electrodes located with respect to the innervation zone ?

There is some confusion about the definition of sensor. Is this an electrode or an electrode pair ?.

The bandwidth of analog and digital notch filters (no capital for notch) also affect sEMG features and should be given.

Did the subject actually performed a movement? Or were the contractions oisometric?

Captions of Fig. 2, 3 and 4 could provide more explanation which movements or efforts were classified or how  classification accuracy is defined.

Specific comments

Line 1 “the quantity and location of the EMG signals” -> “the number and position of sEMG electrodes”

Line 5 “different amounts of EMG signal channels”  -> “different number of sEMG signal channels”

Line 10 give time intervals instead of number of samples

Line 21 “ the sensor is placed under the skin”  ->  “the sensor is intramuscular”

After line 77. MAV: it refers   to the average of the sum of…”  -> MAV: is the average of the N absolute values of the sEMG samples within a given time epoch”

After eq. 4.  “signal change of sign” -> “signal changes sign”

Line 121-122  “the best (no better) signal quality! According to which criterion?. What is the meaning of 66% of muscle length? Starting from where?

Secion 2.2 and 2.3 have the same title.

Eq 8. The symbols in the equation are not defined.

Line 148 “channels shown” -> “channels are shown”

Line 159 “muscles with opposite function” -> “antagonist muscles”

Line 169 “best” according to which crierion?

Line 178 “at the moment of March” ?? not clear. Do you mean “During gait”?

Author Response

1)        The Manuscript ID: electronics-439471 presents some major weaknesses and many minor points mostly related to language. The overall objecives are not clear. While classification of hand movements based on EMG may be important for prosthesis control, why is classification of lower limb movement important?The purpose of the work is to test a classification algorithm that would identify lower limb movements based on a small number of channels of sEMG.  The major objection is that the features of any sEMG channel strongly depend on electrode size, interelectrode distance and electrode location on a muscle.These geometrical parameters a not (or very poorly reported). The issue of repeatability of the classification results when the electrodes are repositioned is not addressed. Should the system be retrained every time it is used? This is not mentioned.

Authors appreciate the comment. The reference Sacco et al., 2009, “A method for better positioning bipolar electrodes for lower limb EMG recordings during dynamic contractions”, Journal of Neuroscience Methods, Vol. 180, pp. 133-137, it shows the position, the opltimal distance of 2.5 cm, and the location of the electrodes, then, we follow the rules in order to have similar results. In the new version of our article, it is mentioned that the electrodes were placed in each subject in just one session.

The authors are referred to the books

1      Hermens H., Freriks B, Merletti R., Stegeman D., Blok J., Rau G., Disselhorst-Klug C., Hagg G., European Recommendations for Surface Electromyography, RRD publish. ISBN 90-75452-15-2, 1999.

2      Barbero M., Rainoldi A, Merletti R. Atlas of muscle innervation zones: understanding surface EMG and its applications, Springer, Italy 2012

3      Merletti R, Farina D. (edts) Surface Electromyography: physiology, engineering and applications,  IEEE Press / J Wiley, USA, May 2016

And to the article :

Afsharipour B., Soedirdjo S Merletti R  Two dimensional surface EMG: the effect of electrode size, interelectrode distance and image truncation. BSPC 2019; 49, 298-307

In particular,  Fig 4 of the latter paper shows the low pass filtering effect of electrode size  and Fig. 9 shows the combined effect of size and interelectrode distance. Both affect ALL the sEMG features selected by the authors. A separation of 25 mm between two electrodes of one pair would create multiple dips in  the differential sEMG signals  (see Fig 9 of the paper mentioned above or fg 2.5 page 40 of  the third book indicated above. How were the electrodes located with respect to the innervation zone ?

Thank you for the recommend the references, we are referencing now in the updated version of our paper.

There is some confusion about the definition of sensor. Is this an electrode or an electrode pair ?

Thanks for the comment. The revision has been attended.

The bandwidth of analog and digital notch filters (no capital for notch) also affect sEMG features and should be given.

In this version of the paper it is mentioned during the introduction the band of rejection regarding the digital filters, also, it is mentioned several references that indicate the improvements that the system has when the frequencies are removed.

Did the subject actually performed a movement? Or were the contractions oisometric?

Thanks for the comment. In section 4 is mentioned that the subject executes certain movemets. Anyway, comment was attended and the phrase was rewritten.

Captions of Fig. 2, 3 and 4 could provide more explanation which movements or efforts were classified or how  classification accuracy is defined.

We appreciate the observation. The definition of accuracy for classification is: to consider the rate of asserts vs. the total.

Specific comments

1      Line 1 “the quantity and location of the EMG signals” -> “the number and position of sEMG electrodes”.

Thanks for the comment. The revision has been attended.

2      Line 5 “different amounts of EMG signal channels”  -> “different number of sEMG signal channels”

Thanks for the comment. The revision has been attended.

3      Line 10 give time intervals instead of number of samples

Thanks for the comment. The revision has been attended.

4      Line 21 “ the sensor is placed under the skin”  ->  “the sensor is intramuscular”

Thanks for the comment. The sensor is not intramuscular.

5      After line 77. MAV: it refers   to the average of the sum of…”  -> MAV: is the average of the N absolute values of the sEMG samples within a given time epoch”

Thanks for the comment. The comment is correct.

6      After eq. 4.  “signal change of sign” -> “signal changes sign”

Thanks for the comment. The revision has been attended.

7      Line 121-122  “the best (no better) signal quality! According to which criterion?. What is the meaning of 66% of muscle length? Starting from where?

The reference Sacco et al., 2009, “A method for better positioning bipolar electrodes for lower limb EMG recordings during dynamic contractions”, Journal of Neuroscience Methods, Vol. 180, pp. 133-137, mention how to measure this signals and the criterions.

8      Secion 2.2 and 2.3 have the same title.

Thanks for the comment. The revision has been attended.

9      Eq 8. The symbols in the equation are not defined.

Thanks for the comment. The revision has been attended.

10   Line 148 “channels shown” -> “channels are shown”

Thanks for the comment. The revision has been attended.

11   Line 159 “muscles with opposite function” -> “antagonist muscles”

Thanks for the comment. The revision has been attended.

12   Line 169 “best” according to which crierion?

Thanks for the comment. We correct the term in order to have a better aproach and remove the term "best".

13   Line 178 “at the moment of March” ?? not clear. Do you mean “During gait”?

Thanks for the comment. Yes, we wanted to say during gait.

Authors appreciate the comment. Then, we improved our work substantially. We performed a statistical analysis using adding a channel forward selection of variables based on the area of ROC (Receiver Operating Characteristic) curve multi-class and a classification error rate. Also, ANOVA was carried out to validate the hypothesis.

Moreover, the number of subjects have been increased to 8 to have sufficient statistical tests. Therefore, the number of samples is 1120. Finally, with the addition of the statistical analysis, it was not considered necessary to keep all the tables in the last version of the article, instead of these, Figure 5 was added to summarize the information.

Reviewer 2 Report

Authors have tried  to classify limb movement using EMG signals. Their method to predict movement with good accuracy is not very clear. I want authors to strengthen their conclusion. They can do couple of things to improve the quality of manuscript - 

provide a comparison table of their algo based classification results with those published in literature. This will give readers a clear picture of how good is their classification method. There are many papers on EMG, EEG, .. signal classification.

I suggest them to increase number of test subjects. It will help to build better statistical results.

Look into how these classifiers behave under noisy environments and provide some results on it. This is very important when measurement is done using surface EMG or EEG. 

    4. I also like to see what future application is possible with this method. 

Author Response

1      Authors have tried to classify limb movement using EMG signals. Their method to predict movement with good accuracy is not very clear. I want authors to strengthen their conclusion. They can do couple of things to improve the quality of manuscript provide a comparison table of their algo based classification results with those published in literature. This will give readers a clear picture of how good is their classification method. There are many papers on EMG, EEG, signal classification. I suggest them to increase number of test subjects. It will help to build better statistical results. Look into how these classifiers behave under noisy environments and provide some results on it. This is very important when measurement is done using surface EMG or EEG.4. I also like to see what future application is possible with this method.

Authors appreciate the comment. In the new version of the paper, a better description was detailed for the procedure and the classification. We did not attempt comparisons with previous classifications. It is because the main idea of the paper is the result launched by variation the number of electrodes instead of the method. In this version of the manuscript, the conclusions were reworked in order to show 4 subjects more that allow us to have improvements in the statistical area.

We performed a statistical analysis using adding a channel forward selection of variables based on the area of ROC (Receiver Operating Characteristic) curve multi-class and a classification error rate. Also, ANOVA was carried out to validate the hypothesis.

Moreover, the number of subjects have been increased to 8 to have sufficient statistical tests. Therefore, the number of samples is 1120.

Reviewer 3 Report

In this study, the authors compare accuracy of the muscle movement type classification when using different amount of data and different number of muscles. They use surface EMG data collected from 4 human subjects from 4 different muscle groups. Then they ask the subjects to make different types of movements and use the support vector machine to classify EMG data back to the different muscle movements. They conclude that while increasing channels from 1 to 4 definitely improves accuracy, 2 versus 3 channel difference is negligible. This is a very interesting and helpful study for the research community. However, I the authors need to address the following concerns.

1.    The number of subjects is too small in this study to make valid conclusions. They authors should increase it.

2.    The authors should perform statistical tests for all the claims they are making in the paper. They are doing multiple comparisons using the data from the same subjects. Hence, they will need to also use a statistical test that accounts for repeated measures. They can consider repeated measures multifactor ANOVA for example. However, they should collect data from more subjects before they attempting to do the statistical tests.

3.    The authors should show a separate plot where they compare the mean accuracy with the different number of channels. Here in the x-axis, you will have 1, 2, 3 and 4 channels and on the y-axis, the average accuracy (average across different combinations) for each channel numbers.

4.    Line 80 – the authors should change “one of correlated” to “one of uncorrelated” as PCA whitens the data or in other words, in the PC space, the data are uncorrelated.

5.    Line 23 – “this former” should be changed to “this later”.

6.    In the Introduction, line 26, the authors should specify what classification they are talking about. They could first mention that they are trying to classify the different types of movements.

7.    Line 145, the authors use the term “significant”. However, this is a statistical term, meaning that, they have to provide a p-value whenever the claim something is significant. Hence, authors should use that term only when they could provide a p-value.

8.    Authors should explain a little bit more about how they selected the “samples” (250,500 and 1000) from the raw data. Did they pick those segments randomly or using some other criteria? 

Author Response

1      In this study, the authors compare accuracy of the muscle movement type classification when using different amount of data and different number of muscles. They use surface EMG data collected from 4 human subjects from 4 different muscle groups. Then they ask the subjects to make different types of movements and use the support vector machine to classify EMG data back to the different muscle movements. They conclude that while increasing channels from 1 to 4 definitely improves accuracy, 2 versus 3 channel difference is negligible. This is a very interesting and helpful study for the research community. However, I the authors need to address the following concerns.

1.     The number of subjects is too small in this study to make valid conclusions. They authors should increase it.

Thanks for the comment. The number of subjects has been increased to 8 to have sufficient statistical tests. Therefore, the number of samples is 1120.

2.     The authors should perform statistical tests for all the claims they are making in the paper. They are doing multiple comparisons using the data from the same subjects. Hence, they will need to also use a statistical test that accounts for repeated measures. They can consider repeated measures multifactor ANOVA for example. However, they should collect data from more subjects before they attempting to do the statistical tests.

Thanks for the comment. We performed a statistical analysis using adding a channel forward selection of variables based on the area of ROC (Receiver Operating Characteristic) curve multi-class and a classification error rate. Also, ANOVA was carried out to validate the hypothesis.

3.     The authors should show a separate plot where they compare the mean accuracy with the different number of channels. Here in the x-axis, you will have 1, 2, 3 and 4 channels and on the y-axis, the average accuracy (average across different combinations) for each channel numbers.

Thanks for the comment. Graph 3 of this new version show the results obtained by 3 and 4 channels and to make a comparison.

According to graph 1, it has just a single channel and graph 2, two channels.

4.     Line 80 – the authors should change “one of correlated” to “one of uncorrelated” as PCA whitens the data or in other words, in the PC space, the data are uncorrelated.

Thanks for the comment. The revision has been attended.

5.     Line 23 – “this former” should be changed to “this later”.

Thanks for the comment. The revision has been attended.

6.     In the Introduction, line 26, the authors should specify what classification they are talking about. They could first mention that they are trying to classify the different types of movements.

Thanks for the comment. The revision has been attended.

7.     Line 145, the authors use the term “significant”. However, this is a statistical term, meaning that, they have to provide a p-value whenever the claim something is significant. Hence, authors should use that term only when they could provide a p-value.

Thanks for the comment. ANOVA was carried out to validate the hypothesis.

8.     Authors should explain a little bit more about how they selected the “samples” (250,500 and 1000) from the raw data. Did they pick those segments randomly or using some other criteria?

Authors appreciate the comment. The window recommended in the literature is 250. However, it was increased the samples to obtain less error.

Round 2

Reviewer 1 Report

The authors corrected only the English errors I pointed out.

There are many more.

The manuscript must have a language revision. Will the journal provide it?

Line 113 – 116.

“Each pair of sensors was placed following the distances described in [29] and separated 2.5 cm

 between them to obtain the better signal quality [29]: for VL the best place is at 66% from muscle

length while for the TA is at 47.5%. The electrode position in GM is at 38% of the muscle length and for

BF 50% presents the better quality of the signal.”

Please indicate where the measurement of distance start from, as requested in the first review.

Please discuss the fact that an interelectrode distance of 25 mm introduces major aliasing. This value is critical as well as the electrode size. References to this issue have not been included and the issue is not discussed.

See paper by Afsharipour et al on BSPC, mentioned in the previous review.

Please provide the electrode dimension. The reader must be able to reproduce the experiments. DO NOT force the reader to look up other papers for trivial information.

The answer to the issue raised in the first review is NOT satisfactory.

Line 136 “started from a relaxation state and then performed the movement and holding it for 5 seconds”.

Not clear. the final position was held for 5 s? Were the EMG signals acquired during these five seconds in isometric conditions or during the movement? Please clarify.

Line 150  

 “Table 1. Better accuracy results obtained.”  Better than what ? Do you mean best? If so, best among what? Perhaps the title should be: results obtained with the highest accuracy ?   Among how many?  Please expand the title of table 1 and 2 and explain the table content better.

Line 155 “The results in Table 2 are the lowest scores, and these show that tibialis anterior has not enough

 information to make a good classification, even if it is combined with the TA muscle.” I do not understand this sentence.  Tibialis anterior and TA are the same muscle….

Author Response

Reviewer 1

1)        The authors corrected only the English errors I pointed out. There are many more. The manuscript must have a language revision. Will the journal provide it?

The language of the manuscript has now been checked by native English speakers.

2)        Lines 113 – 116.

“Each pair of sensors was placed following the distances described in [29] and separated 2.5 cm between them to obtain the better signal quality [29]: for VL the best place is at 66% from muscle length while for the TA is at 47.5%. The electrode position in GM is at 38% of the muscle length and for BF 50% presents the better quality of the signal.”

Please indicate where the measurement of distance start from, as requested in the first review.

Please discuss the fact that an interelectrode distance of 25 mm introduces major aliasing. This value is critical as well as the electrode size. References to this issue have not been included and the issue is not discussed.

See paper by Afsharipour et al on BSPC, mentioned in the previous review.

Please provide the electrode dimension. The reader must be able to reproduce the experiments. DO NOT force the reader to look up other papers for trivial information.

The answer to the issue raised in the first review is NOT satisfactory.

Authors appreciate the comment. The information requested is in reference [29], and that is the reason why we just paraphrase. Moreover, we took the information of [29] and then, now it is added in the manuscript. We added the cite [30] of the relevant work by Afsharipour et al. published in BSPC.

3)        Line 136 “started from a relaxation state and then performed the movement and holding it for 5 seconds”.

Not clear. the final position was held for 5 s? Were the EMG signals acquired during these five seconds in isometric conditions or during the movement? Please clarify.

Authors appreciate the comment, we re-write the phrase.

“In the experiment, the subjects were sitting and started from a relaxation state and then performed the movement and held it for 5 seconds, and then they returned to the relaxation position.”

4)        Line 150

“Table 1. Better accuracy results obtained.”  Better than what ? Do you mean best? If so, best among what? Perhaps the title should be: results obtained with the highest accuracy ?   Among how many?  Please expand the title of table 1 and 2 and explain the table content better.

Authors appreciate the comment, we re-write the title.

“Best accuracy results obtained among the eight subjects.

5)        Line 155 “The results in Table 2 are the lowest scores, and these show that tibialis anterior has not enough  information to make a good classification, even if it is combined with the TA muscle.” I do not understand this sentence.  Tibialis anterior and TA are the same muscle….

Authors appreciate the observation, we correct the phrase, “The results in Table 2 are the lowest scores, and these show that tibialis anterior has not enough information to make a good classification, even if it is combined with the VL muscle.”

Reviewer 2 Report

Authors have addressed my review questions. I suggest publication of this manuscript at the earliest possible. 

Author Response

Thanks for your comment. We ask for English corrections with a native speaker.